# Dimensionality Reduction of Massive Sparse Datasets Using Coresets

**Dan Feldman**
University of Haifa
Haifa, Israel
dannyf.post@gmail.com

**Mikhail Volkov**
CSAIL, MIT
Cambridge, MA, USA
mikhail@csail.mit.edu

**Daniela Rus**
CSAIL, MIT
Cambridge, MA, USA
rus@csail.mit.edu

## Abstract

In this paper we present a practical solution with performance guarantees to the problem of dimensionality reduction for very large scale sparse matrices. We show applications of our approach to computing the Principle Component Analysis (PCA) of any $n \times d$ matrix, using one pass over the stream of its rows. Our solution uses coresets: a scaled subset of the $n$ rows that approximates their sum of squared distances to *every* $k$-dimensional *affine* subspace. An open theoretical problem has been to compute such a coreset that is independent of both $n$ and $d$. An open practical problem has been to compute a non-trivial approximation to the PCA of very large but sparse databases such as the Wikipedia document-term matrix in a reasonable time. We answer both of these questions affirmatively. Our main technical result is a new framework for deterministic coreset constructions based on a reduction to the problem of counting items in a stream.

## 1 Introduction

Algorithms for dimensionality reduction usually aim to project an input set of $d$-dimensional vectors (database records) onto a $k \leq d - 1$ dimensional affine subspace that minimizes the sum of squared distances to these vectors, under some constraints. Special cases include the Principle Component Analysis (PCA), Linear regression ($k = d - 1$), Low-rank approximation ($k$-SVD), Latent Drichlet Analysis (LDA) and Non-negative matrix factorization (NNMF). Learning algorithms such as $k$-means clustering can then be applied on the low-dimensional data to obtain fast approximations with provable guarantees. To our knowledge, unlike SVD, there are no algorithms or coreset constructions with performance guarantees for computing the PCA of sparse $n \times n$ matrices in the streaming model, i.e. using memory that is poly-logarithmic in $n$. Much of the large scale high-dimensional data sets available today (e.g. image streams, text streams, etc.) are sparse. For example, consider the text case of Wikipedia. We can associate a matrix with Wikipedia, where the English words define the columns (approximately 1.4 million) and the individual documents define the rows (approximately 4.4 million documents). This large scale matrix is sparse because most English words do not appear in most documents. The size of this matrix is huge and no existing dimensionality reduction algorithm can compute its eigenvectors. To this point, running the state of the art SVD implementation from GenSim on the Wikipedia document-term matrix crashes the computer very quickly after applying its step of random projection on the first few thousand documents. This is because such dense vectors, each of length 1.4 million, use all of the computer's RAM capacity.

In this paper we present a dimensionality reduction algorithms that can handle very large scale sparse data sets such as Wikipedia and returns provably correct results. A long-open research question has been whether we can have a coreset for PCA that is both small in size and a subset of the original data. In this paper we answer this question affirmatively and provide an efficient construction. We also show that this algorithm provides a practical solution to a long-standing open practical problem: computing the PCA of large matrices such as those associated with Wikipedia.

## 2 Problem Formulation

Given a matrix $A$, a coreset $C$ in this paper is defined as a weighted subset of rows of $A$ such that the sum of squared distances from any given $k$-dimensional subspace to the rows of $A$ is approximately the same as the sum of squared weighted distances to the rows in $C$. Formally,

For a compact set $S \in \mathbb{R}^d$ and a vector $x$ in $\mathbb{R}^d$, we denote the Euclidean distance between $x$ and its closest points in $S$ by

$$\operatorname{dist}^2(x, S) := \min_{s \in S} \|x - s\|_2^2$$

For an $n \times d$ matrix $A$ whose rows are $a_1, \ldots, a_n$, we define the sum of the squared distances from $A$ to $S$ by

$$\operatorname{dist}^2(A, S) := \sum_{i=1}^n \operatorname{dist}^2(a_i, S)$$

**Definition 1** (($k, \varepsilon$)-coreset). *Given a $n \times d$ matrix $A$ whose rows $a_1, \cdots, a_n$ are $n$ points (vectors) in $\mathbb{R}^d$, an error parameter $\varepsilon \in (0, 1]$, and an integer $k \in [1, d-1] = \{1, \cdots, d-1\}$ that represents the desired dimensionality reduction, $n$ ($k, \varepsilon$)-coreset for $A$ is a weighted subset $C = \{w_i a_i \mid w_i > 0 \text{ and } i \in [n]\}$ of the rows of $A$, where $w = (w_1, \cdots, w_n) \in [0, \infty)^n$ is a non-negative weight vector, such that for every affine $k$-subspace $S$ in $\mathbb{R}^d$ we have*

$$\left| \operatorname{dist}^2(A, S)) - \operatorname{dist}^2(C, S)) \right| \leq \varepsilon \operatorname{dist}^2(A, S)). \tag{1}$$

That is, the sum of squared distances from the $n$ points to $S$ approximates the sum of squared weighted distances $\sum_{i=1}^n w_i^2 (\operatorname{dist}(a_i, S))^2$ to $S$. The approximation is up to a multiplicative factor of $1 \pm \varepsilon$. By choosing $w = (1, \cdots, 1)$ we obtain a trivial ($k, 0$)-coreset. However, in a more efficient coreset most of the weights will be zero and the corresponding rows in $A$ can be discarded. The cardinality of the coreset is thus the sparsity of $w$, given by $|C| = \|w\|_0 := |\{w_i \neq 0 \mid i \in [n]\}|$.

If $C$ is small, then the computation is efficient. Because $C$ is a weighted subset of the rows of $A$, if $A$ is sparse, then $C$ is also sparse. A long-open research question has been whether we can have such a coreset that is both *of size* independent of the input dimension ($n$ and $d$) and a *subset of the original input rows*.

### 2.1 Related Work

In [24] it was recently proved that an ($k, \varepsilon$) coreset of size $|C| = O(dk^3/\varepsilon^2)$ exists for every input matrix, and distances to the power of $z \geq 1$ where $z$ is constant. The proof is based on a general framework for constructing different kinds of coresets, and is known as *sensitivity* [10, 17]. This coreset is efficient for tall matrices, since its cardinality is independent of $n$. However, it is useless for "fat" or square matrices (such as the Wikipedia matrix above), where $d$ is in the order of $n$, which is the main motivation for our paper. In [5], the Frank-Wolfe algorithm was used to construct different types of coresets than ours, and for different problems. Our approach is based on a solution that we give to an open problem in [5], however we can see how it can be used to compute the coresets in [5] and vice versa. For the special case $z = 2$ (sum of squared distances), a coreset of size $O(k/\varepsilon^2)$ was suggested in [7] with a randomized version in [8] for a stream of $n$ points that, unlike the standard approach of using merge-and-reduce trees, returns a coreset of size independent of $n$ with a constant probability. These result minimizes the $\| \cdot \|_2$ error, while our result minimizes the Frobenius norm, which is always higher, and may be higher by a factor of $d$. After appropriate weighting, we can apply the uniform sampling of size $O(k/\varepsilon^2)$ to get a coreset with a small Frobenius error [14], as in our paper. However, in this case the probability of success is only constant. Since in the streaming case we compute roughly $n$ coresets (formally, $O(n/m)$ coresets, where $m$ is the size of the coreset) the probability that all these coresets constructions will succeed

is close to zero (roughly $1/n$). Since the probability of failure in [14] reduces linearly with the size of the coreset, getting a constant probability of success in the streaming model for $O(n)$ coresets would require to take coresets of size that is no smaller than the input size.

There are many papers, especially in recent years, regarding data compression for computing the SVD of large matrices. None of these works addresses the fundamental problem of computing a sparse approximated PCA for a large matrix (in both rows and columns), such as Wikipedia. The reason is that current results use sketches which do no preserve the sparsity of the data (e.g. because of using random projections). Hence, neither the sketch nor the PCA computed on the sketch is sparse. On the other side, we define coreset as a small weighted subset of rows, which is thus sparse if the input is sparse. Moreover, the low rank approximation of a coreset is sparse, since each of its right singular vectors is a sum of a small set of sparse vectors. While there are coresets constructions as defined in this paper, all of them have cardinality of at least $d$ points, which makes them impractical for large data matrices, where $d \geq n$. In what follows we describe these recent results in details.

The recent results in [7, 8] suggest coresets that are similar to our definition of coresets (i.e., weighted subsets), and do preserve sparsity. However, as mentioned above they minimize the 2-norm error and not the larger Frobesnius error, and maybe more important, they provide coresets for $k$-SVD (i.e., $k$-dimensional subspaces) and not for PCA ($k$-dimensional affine subspaces that might not intersect the origin). In addition [8] works with constant probability, while our algorithm is deterministic (works with probability 1).

**Software.** Popular software for computing SVD such as GenSim [21], redsvd [12] or the MATLAB sparse SVD function (`svds`) use sketches and crash for inputs of a few thousand of documents and a dimensionality reduction (approximation rank) $k < 100$ on a regular laptop, as expected from the analysis of their algorithms. This is why existing implementations (including Gensim) extract topics from large matrices (e.g. Wikipedia), based on low-rank approximation of only small subset of few thousands of selected words (matrix columns), and not the complete Wikipedia matrix.Even for $k = 3$, running the implementation of sparse SVD in Hadoop [23] took several days [13]. Next we give a broad overview of the very latest state of the dimensionality reduction methods, such as the Lanczoz algorithm [16] for large matrices, that such systems employ under the hood.

**Coresets.** Following a decade of research in [24] it was recently proved that an $(\varepsilon, k)$-coreset for low rank approximation of size $|C| = O(dk^3/\varepsilon^2)$ exists for every input matrix. The proof is based on a general framework for constructing different kinds of coresets, and is known as *sensitivity* [10, 17]. This coreset is efficient for tall matrices, since its cardinality is independent of $n$. However, it is useless for "fat" or square matrices (such as the Wikipedia matrix above), where $d$ is in the order of $n$, which is the main motivation for our paper. In [5], the Frank-Wolfe algorithm was used to construct different types of coresets than ours, and for different problems. Our approach is based on a solution that we give to an open problem in [5].

**Sketches.** A *sketch* in the context of matrices is a set of vectors $u_1, \cdots, u_s$ in $\mathbb{R}^d$ such that the sum of squared distances $\sum_{i=1}^{n}(\mathrm{dist}(a_i, S))^2$ from the input $n$ points to *every* $k$-dimensional subspace $S$ in $\mathbb{R}^d$, can be approximated by $\sum_{i=1}^{n}(\mathrm{dist}(u_i, S))^2$ up to a multiplicative factor of $1 \pm \varepsilon$. Note that even if the input vectors $a_1, \cdots, a_n$ are sparse, the sketched vectors $u_1, \cdots, u_s$ in general are not sparse, unlike the case of coresets. A sketch of cardinality $d$ can be constructed with no approximation error ($\varepsilon = 0$), by defining $u_1, \cdots, u_d$ to be the $d$ rows of the matrix $DV^T$ where $UDV^T = A$ is the SVD of $A$. It was proved in [11] that taking the first $O(k/\varepsilon)$ rows of $DV^T$ yields such a sketch, i.e. of size independent of $n$ and $d$.

The first sketch for sparse matrices was suggested in [6], but like more recent results, it assumes that the complete matrix fits in memory. Other sketching methods that usually do not support streaming include random projections [2, 1, 9] and randomly combined rows [20, 25, 22, 18].

**The Lanczoz Algorithm.** The Lanczoz method [19] and its variant [15] multiply a large matrix by a vector for a few iterations to get its largest eigenvector $v_1$. Then the computation is done recursively after projecting the matrix on the hyperplane that is orthogonal to $v_1$. However, $v_1$ is in general not sparse even $A$ is sparse. Hence, when we project $A$ on the orthogonal subspace to $v_1$, the resulting matrix is dense for the rest of the computations ($k > 1$). Indeed, our experimental results show that the MATLAB `svds` function which uses this method runs faster than the exact SVD, but crashes on large input, even for small $k$.

This paper builds on this extensive body of prior work in dimensionality reduction, and our approach uses coresets to solve the time and space challenges.

## 2.2 Key Contributions

Our main result is the first algorithm for computing an $(k, \varepsilon)$-coreset $C$ of size independent of both $n$ and $d$, for any given $n \times d$ input matrix. The algorithm takes as input a finite set of $d$-dimensional vectors, a desired approximation error $\varepsilon$, and an integer $k \geq 0$. It returns a weighted subset $S$ (coreset) of $k^2/\varepsilon^2$ such vectors. This coreset $S$ can be used to approximate the sum of squared distances from the matrix $A \in \mathbb{R}^{n \times d}$, whose rows are the $n$ vectors seen so far, to any $k$-dimensional affine subspace in $\mathbb{R}^d$, up to a factor of $1 \pm \varepsilon$. For a (possibly unbounded) stream of such input vectors the coreset can be maintained at the cost of an additional factor of $\log^2 n$.

The polynomial dependency on $d$ of the cardinality of previous coresets made them impractical for fat or square input matrices, such as Wikipedia, images in a sparse feature space representation, or adjacency matrix of a graph. If each row of in input matrix $A$ has $O(\text{nnz})$ non-zeroes entries, then the update time per insertion, the overall memory that is used by our algorithm, and the low rank approximation of the coreset $S$ is $O(\text{nnz} \cdot k^2/\varepsilon^2)$, i.e. independent of $n$ and $d$.

We implemented our algorithm to obtain a low-rank approximation for the term-document matrix of Wikipedia with provable error bounds. Since our streaming algorithm is also "embarrassingly parallel" we run it on Amazon Cloud, and receive a significantly better running time and accuracy compared to existing heuristics (e.g. Hadoop/MapReduce) that yield non-sparse solutions.

The key contributions in this work are:

1. A new algorithm for dimensionality reduction of sparse data that uses a weighted subset of the data, and is independent of both the size and dimensionality of the data.
2. An efficient algorithm for computing such a reduction, with provable bounds on size and running time (cf. `http://people.csail.mit.edu/mikhail/NIPS2016`).
3. A system that implements this dimensionality reduction algorithm and an application of the system to compute latent semantic analysis (LSA) of the entire English Wikipedia.

# 3 Technical Solution

Given a $n \times d$ matrix $A$, we propose a construction mechanism for a matrix $C$ of size $|C| = O(k^2/\varepsilon^2)$ and claim that it is a $(k, \varepsilon)$-coreset for $A$. We use the following corollary for Definition 1 of a coreset, based on simple linear algebra that follows from the geometrical definitions (e.g. see [11]).

**Property 1** (Coreset for sparse matrix). *Let $A \in \mathbb{R}^{n \times d}$, $k \in [1, d-1]$ be an integer, and let $\varepsilon > 0$ be an error parameter. For a diagonal matrix $W \in \mathbb{R}^{n \times n}$, the matrix $C = WA$ is a $(k, \varepsilon)$-coreset for $A$ if for every matrix $X \in \mathbb{R}^{d \times (d-k)}$ such that $X^T X = I$, we have*

$$\text{(i)} \quad \left| 1 - \frac{\|WAX\|}{\|AX\|} \right| \leq \varepsilon, \quad \text{and} \quad \text{(ii)} \quad \|A - WA\| < \varepsilon \operatorname{var}(A) \tag{2}$$

*where $\operatorname{var}(A)$ is the sum of squared distances from the rows of $A$ to their mean.*

The goal of this paper is to prove that such a coreset (Definition 1) exists for any matrix $A$ (Property 1) and can be computed efficiently. Formally,

**Theorem 1.** *For every input matrix $A \in \mathbb{R}^{n \times d}$, an error $\varepsilon \in (0, 1]$ and an integer $k \in [1, d-1]$:*

(a) *there is a $(k, \varepsilon)$-coreset $C$ of size $|C| = O(k^2/\varepsilon^2)$;*

(b) *such a coreset can be constructed in $O(k^2/\varepsilon^2)$ time.*

Theorem 1 is the formal statement for the main technical contribution of this paper. Sections 3–5 constitute a proof for Theorem 1.

To establish Theorem 1(a), we first state our two main results (Theorems 2 and 3) axiomatically, and show how they combine such that Property 1 holds. Thereafter we prove the these results in Sections 4 and 5, respectively. To prove Theorem 1(b) (efficient construction) we present an algorithm for

**Algorithm 1** CORESET-SUMVECS$(A, \varepsilon)$

1: **Input:** $A$: $n$ input points $a_1, \ldots, a_n$ in $\mathbb{R}^d$
2: **Input:** $\varepsilon \in (0, 1)$: the approximation error
3: **Output:** $w \in [0, \infty)^n$: non-negative weights
4: $A \leftarrow A - \text{mean}(A)$
5: $A \leftarrow c\,A$ where $c$ is a constant s.t. $\text{var}(A) = 1$
6: $w \leftarrow (1, 0, \ldots, 0)$
7: $j \leftarrow 1, \ p \leftarrow A_j, \ J \leftarrow \{j\}$
8: $M_j = \{y^2 \mid y = A \cdot A_j^T\}$
9: **for** $i = 1, \ldots, n$ **do**
10: $\quad j \leftarrow \text{argmin}\, \{w_J \cdot M_J\}$
11: $\quad G \leftarrow W' \cdot A_J$ where $W'_{i,i} = \sqrt{w_i}$
12: $\quad \|c\| = \|G^T G\|_F^2$
13: $\quad c \cdot p = \sum_{i=1}^{|J|} G\, p^T$
14: $\quad \|c - p\| = \sqrt{1 + \|c\|^2 - c \cdot p}$
15: $\quad \text{comp}_p(v) = 1/\|c - p\| - (c \cdot p)/\|c - p\|$
16: $\quad \|c - c'\| = \|c - p\| - \text{comp}_p(v)$
17: $\quad \alpha = \|c - c'\|/\|c - p\|$
18: $\quad w \leftarrow w(1 - |\alpha|)$
19: $\quad w_j \leftarrow w_j + \alpha$
20: $\quad w \leftarrow w/\sum_{i=1}^n w_i$
21: $\quad M_j \leftarrow \{y^2 \mid y = A \cdot A_j^T\}$
22: $\quad J \leftarrow J \cup \{j\}$
23: $\quad$ **if** $\|c\|^2 \le \varepsilon$ **then**
24: $\quad\quad$ **break**
25: $\quad$ **end if**
26: **end for**
27: **return** $w$

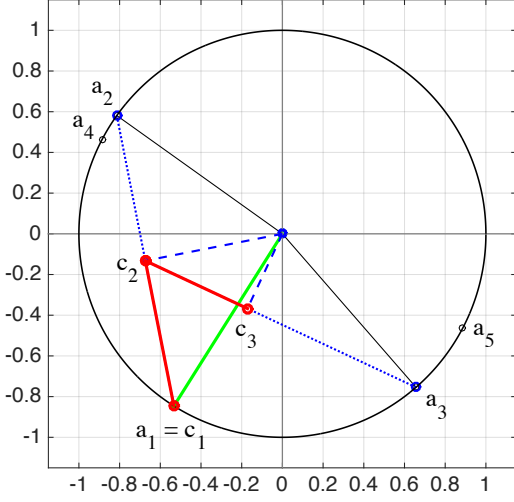

(a) Coreset for sum of vectors algorithm | (b) Illustration showing first 3 steps of the computation

computing a matrix $C$, and analyze the running time to show that the $C$ can be constructed in $O(k^2/\varepsilon^2)$ iterations.

Let $A \in \mathbb{R}^{n \times d}$ be a matrix of rank $d$, and let $U\Sigma V^T = A$ denote its full SVD. Let $W \in \mathbb{R}^{n \times n}$ be a diagonal matrix. Let $k \in [1, d-1]$ be an integer. For every $i \in [n]$ let

$$v_i = \left( U_{i,1}, \cdots, U_{i,k}, \frac{U_{i,k+1:d}\Sigma_{k+1:d,k+1:d}}{\|\Sigma_{k+1:d,k+1:d}\|}, 1 \right). \tag{3}$$

Then the following two results hold:

**Theorem 2** (Coreset for sum of vectors). *For every set of of $n$ vectors $v_1, \cdots, v_n$ in $\mathbb{R}^d$ and every $\varepsilon \in (0, 1)$, a weight vector $w \in (0, \infty)^n$ of sparsity $\|w\|_0 \le 1/\varepsilon^2$ can be computed deterministically in $O(nd/\varepsilon)$ time such that*

$$\left\| \sum_{i=1}^n v_i - \sum_{i=1}^n w_i v_i \right\| \le \varepsilon \sum_{i=1}^n \|v_i\|^2. \tag{4}$$

Section 4 establishes a proof for Theorem 2.

**Theorem 3** (Coreset for Low rank approximation). *For every $X \in \mathbb{R}^{d \times (d-k)}$ such that $X^T X = I$,*

$$\left| 1 - \frac{\|WAX\|^2}{\|AX\|^2} \right| \le 5 \left\| \sum_{i=1}^n v_i v_i^T - W_{i,i} v_i v_i^T \right\|. \tag{5}$$

Section 5 establishes a proof for Theorem 3.

### 3.1 Proof of Theorem 1

*Proof of Theorem 1(a).* Replacing $v_i$ with $v_i v_i^T$, $\|v_i\|^2$ with $\|v_i v_i^T\|$, and $\varepsilon$ by $\varepsilon/(5k)$ in Theorem 2 yields

$$\left\| \sum_i v_i v_i^T - W_{i,i} v_i v_i^T \right\| \le (\varepsilon/5k) \sum_{i=1}^n \|v_i v_i^T\| = \varepsilon.$$

Combining this inequality with (4) gives

$$\left| 1 - \frac{\|WAX\|^2}{\|AX\|^2} \right| \le 5 \left\| \sum_{i=1}^{n} v_i v_i^T - W_{i,i} v_i v_i^T \right\| \le \varepsilon.$$

Thus the left-most term is bounded by the right-most term, which proves (2). This also means that $C = WA$ is a coreset for $k$-SVD, i.e., (non-affine) $k$-dimensional subspaces. To support PCA (affine subspaces) the coreset $C = WA$ needs to satisfy the expression in the last line of Property 1 regarding its mean. This holds using the last entry (one) in the definition of $v_i$ (3), which implies that the sum of the rows is preserved as in equation (4). Therefore Property 1 holds for $C = WA$, which proves Theorem 1(a).

Claim Theorem 1(b) follows from simple analysis of Algorithm 2 that implements this construction. □

## 4 Coreset for Sum of Vectors ($k = 0$)

In order to prove the general result Theorem 1(a), that is the existence of a $(k, \varepsilon)$-coreset for any $k \in [1, d-1]$, we first establish the special case for $k = 0$. In this section, we prove Theorem 2 by providing an algorithm for constructing a small weighted subset of points that constitutes a general approximation for the sum of vectors.

To this end, we first introduce an intermediate result that shows that given $n$ points on the unit ball with weight distribution $z$, there exists a small subset of points whose weighted mean is approximately the same as the weighted mean of the original points.

Let $D^n$ denote the union over every vector $z \in [0,1]^n$ that represent a distribution, i.e., $\sum_i z_i = 1$. Our first technical result is that for any finite set of unit vectors $a_1, \ldots, a_n$ in $\mathbb{R}^d$, any distribution $z \in D^n$, and every $\varepsilon \in (0,1]$, we can compute a sparse weight vector $w \in D^n$ of sparsity (non-zeroes entries) $\|w\|_0 \le 1/\varepsilon^2$.

**Lemma 1.** *Let $z \in D^n$ be a distribution over $n$ unit vectors $a_1, \cdots, a_n$ in $\mathbb{R}^d$. For $\varepsilon \in (0,1)$, a sparse weight vector $w \in D^n$ of sparsity $s \le 1/\varepsilon^2$ can be computed in $O(nd/\varepsilon^2)$ time such that*

$$\left\| \sum_{i=1}^{n} z_i \cdot a_i - \sum_{i=2}^{n} w_i\, a_i \right\|_2 \le \varepsilon. \tag{6}$$

*Proof of Lemma* 1. Please see Supplementary Material, Section A. □

We prove Theorem 2 by providing a computation of such a sparse weight vector $w$. The intuition for this computation is as follows. Given $n$ input points $a_1, \ldots, a_n$ in $\mathbb{R}^d$, with weighted mean $\sum_i z_i\, a_i = \mathbf{0}$, we project all the points on the unit sphere. Pick an arbitrary starting point $a_1 = c_1$. At each step find the farthest point $a_{j+1}$ from $c_j$, and compute $c_{j+1}$ by projecting the origin onto the line segment $[c_j, a_{j+1}]$. Repeat this for $j = 1, \ldots, N$ iterations, where $N = 1/\varepsilon^2$. We prove that $\|c_i\|^2 = 1/i$, thus if we iterate $1/\epsilon^2$ times, this norm will be $\|c_{1/\epsilon^2}\| = \epsilon^2$. The resulting points $c_i$ are a weighted linear combination of a small subset of the input points. The output weight vector $w \in D^n$ satisfies $c_N = \sum_{i=1}^{n} w_i\, a_i$, and this weighted subset forms the coreset.

Fig. 1a contains the pseudocode for Algorithm 1. Fig. 1b illustrates the first steps of the main computation (lines 9–26). Please see Supplementary Material, Section C for a complete line-by-line analysis of Algorithm 1.

*Proof of Theorem* 2. The proof of Theorem 2 follows by applying Lemma 1 after normalization of the input points and then post-processing the output. □

## 5 Coreset for Low Rank Approximation ($k > 0$)

In Section 4 we presented a new coreset construction for approximating the sum of vectors, showing that given $n$ points on the unit ball there exists a small weighted subset of points that is a coreset for those points. In this section we describe the reduction of Algorithm 1 for $k = 0$ to an efficient algorithm for any low rank approximation with $k \in [1, d-1]$.

**Algorithm 2** CORESET-LOWRANK$(A, k, \varepsilon)$

| | |
|---|---|
| 1: **Input:** $A$: A sparse $n \times d$ matrix | 12: **for** $i = 1, \ldots, \lceil k^2/\varepsilon^2 \rceil$ **do** |
| 2: **Input:** $k \in \mathbb{Z}_{>0}$: the approximation rank | 13: $\quad j \leftarrow \operatorname{argmin}_{i=1,\ldots,n}\{wXX_i\}$ |
| 3: **Input:** $\varepsilon \in \left(0, \frac{1}{2}\right)$: the approximation error | 14: $\quad a = \sum_{i=1}^{n} w_i (X_i^T X_j)^2$ |
| 4: **Output:** $w \in [0, \infty)^n$: non-negative weights | 15: $\quad b = \dfrac{1 - \|PX_j\|_F^2 + \sum_{i=1}^{n} w_i \|PX_i\|_F^2}{\|P\|_F^2}$ |
| 5: Compute $U\Sigma V^T = A$, the SVD of $A$ | |
| 6: $R \leftarrow \Sigma_{k+1:d, k+1:d}$ | 16: $\quad c = \|wX\|_F^2$ |
| 7: $P \leftarrow$ matrix whose $i$-th row $\forall i \in [n]$ is | 17: $\quad \alpha = (1 - a + b) / (1 + c - 2a)$ |
| 8: $\quad P_i = (U_{i,1:k}, U_{i,k+1:d} \cdot \frac{R}{\|R\|_F})$ | 18: $\quad w \leftarrow (1 - \alpha)I_j + \alpha w$ |
| 9: $X \leftarrow$ matrix whose $i$-th row $\forall i \in [n]$ is | 19: **end for** |
| 10: $\quad X_i = P_i / \|P_i\|_F$ | 20: **return** $w$ |
| 11: $w \leftarrow (1, 0, \ldots, 0)$ | |
| (a) 1/2: Initialization | (b) 2/2: Computation |

Conceptually, we achieve this reduction in two steps. The first step is to show that Algorithm 1 can be reduced to an inefficient computation for low rank approximation for matrices. To this end, we first prove Theorem 3, thus completing the existence clause Theorem 1(a).

*Proof of Theorem* 3. Let $\varepsilon = \|\sum_{i=1}^{n}(1 - W_{i,i}^2)v_i v_i^T\|$. For every $i \in [n]$ let $t_i = 1 - W_{i,i}^2$. Set $X \in \mathbb{R}^{d \times (d-k)}$ such that $X^T X = I$. Without loss of generality we assume $V^T = I$, i.e. $A = U\Sigma$, otherwise we replace $X$ by $V^T X$. It thus suffices to prove that $\left|\sum_i t_i \|A_{i,:}X\|^2\right| \leq 5\varepsilon \|AX\|^2$. Using the triangle inequality, we get

$$\left|\sum_i t_i \|A_{i,:}X\|^2\right| \leq \left|\sum_i t_i \|A_{i,:}X\|^2 - \sum_i t_i \|(A_{i,1:k}, \mathbf{0})X\|^2\right| \tag{7}$$

$$+ \left|\sum_i t_i \|(A_{i,1:k}, \mathbf{0})X\|^2\right|. \tag{8}$$

We complete the proof by deriving bounds on (7) and (8), thus proving (5). For the complete proof, please see Supplementary Material, Section B. □

Together, Theorems 2 and 3 show that the error of the coreset is a $1 \pm \varepsilon$ approximation to the true weighted mean. By Theorem 3, we can now simply apply Algorithm 1 to the right hand side of (5) to compute the reduction. The intuition for this inefficient reduction is as follows. We first compute the outer product of each row vector $x$ in the input matrix $A \in \mathbb{R}^{[n \times d]}$. Each such outer products $x^T x$ is a matrix in $\mathbb{R}^{d \times d}$. Next, we expand every such matrix into a vector, in $\mathbb{R}^{d^2}$ by concatenating its entries. Finally, we combine each such vector back to be a vector in the matrix $P \in \mathbb{R}^{n \times d^2}$. At this point the reduction is complete, however it is clear that this matrix expansion is inefficient.

The second step of the reduction is to transform the slow computation of running Algorithm 1 on the expanded matrix $P \in \mathbb{R}^{n \times d^2}$ into an equivalent and provably fast computation on the original set of points $A \in \mathbb{R}^d$. To this end we make use of the fact that each row of $P$ is a sparse vector in $\mathbb{R}^d$ to implicitly run the computation in the original row space $\mathbb{R}^d$. We present Algorithm 2 and prove that it returns the weight vector $w = (w_1, \cdots, w_n)$ of a $(k, \varepsilon)$-coreset for low-rank approximation of the input point set $P$, and that this coreset is small, namely, only $O(k^2/\varepsilon^2)$ of the weights (entries) in $w$ are non-zeros. Fig. 5 contains the pseudocode for Algorithm 2. Please see Supplementary Material, Section D for a complete line-by-line analysis of Algorithm 2.

## 6 Evaluation and Experimental Results

The coreset construction algorithm described in Section 5 was implemented in MATLAB. We make use of the redsvd package [12] to improve performance, but it is not required to run the system. We evaluate our system on two types of data: synthetic data generated with carefully controlled parameters, and real data from the English Wikipedia under the "bag of words" (BOW) model. Synthetic data provides ground-truth to evaluate the quality, efficiency, and scalability of our system, while the Wikipedia data provides us with a grand challenge for latent semantic analysis computation.

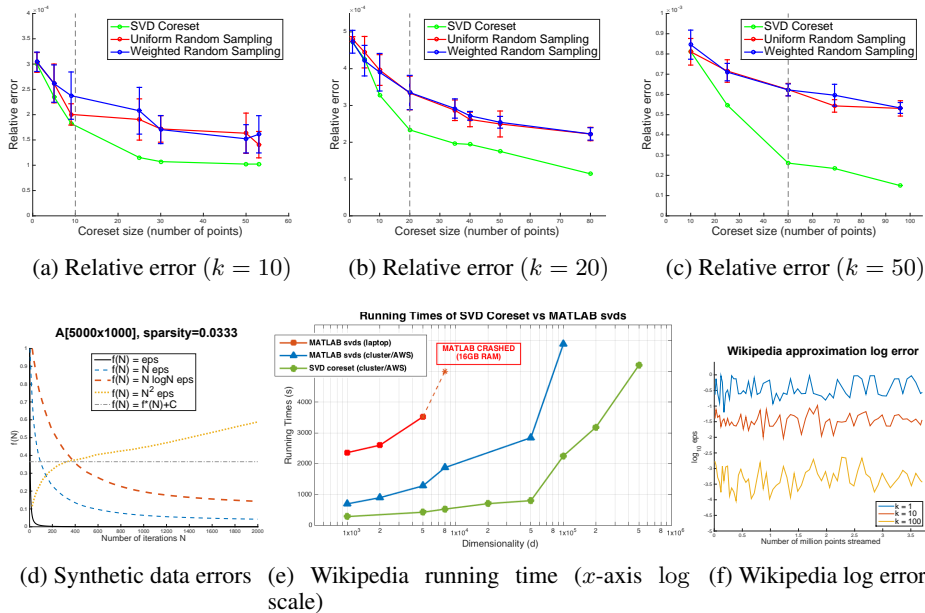

(a) Relative error ($k = 10$)  (b) Relative error ($k = 20$)  (c) Relative error ($k = 50$)

(d) Synthetic data errors  (e) Wikipedia running time ($x$-axis log scale)  (f) Wikipedia log errors

Figure 1: Experimental results for synthetic data (Fig. 1a–1d) and Wikipedia (Fig. 1e–Fig. 1f).

For our synthetic data experiments, we used a moderate size sparse input of ($5000 \times 1000$) to evaluate the relationship between the error $\varepsilon$ and the number of iterations of the algorithm $N$. We then compare our coreset against uniform sampling and weighted random sampling using the squared norms of $U$ ($A = U\Sigma V^T$) as the weights. Finally, we evaluate the efficiency of our algorithm by comparing the running time against the MATLAB `svds` function and against the most recent state of the art dimensionality reduction algorithm [8]. Figure 1a–1d show the exerimental results. Please see Supplementary Material, Section E for a complete description of the experiments.

## 6.1 Latent Semantic Analysis of Wikipedia

For our large-scale grand challenge experiment, we apply our algorithm for computing Latent Semantic Analysis (LSA) on the entire English Wikipedia. The size of the data is $n = 3.69$M (documents) with a dimensionality $d = 7.96$M (words). We specify a nominal error of $\varepsilon = 0.5$, which is a theoretical upper bound for $N = 2k/\varepsilon$ iterations, and show that the coreset error remains bounded. Figure 1f shows the log approximation error, i.e. sum of squared distances of the coreset to the subspace for increasing approximation rank $k = 1, 10, 100$. We see that the log error is proportional to $k$, and as the number of streamed points increases into the millions, coreset error remains bounded by $k$. Figure 1e shows the running time of our algorithm compared against `svds` for increasing dimensionality $d$ and a fixed input size $n = 3.69$M (number of documents).

Finally, we show that our coreset can be used to create a topic model of 100 topics for the entire English Wikipedia. We construct the coreset of size $N = 1000$ words. Then to generate the topics, we compute a projection of the coreset onto a subspace of rank $k = 100$. Please see Supplementary Material, Section F for more details, including an example of the topics obtained in our experiments.

## 7 Conclusion

We present a new approach for dimensionality reduction using coresets. Our solution is general and can be used to project spaces of dimension $d$ to subspaces of dimension $k < d$. The key feature of our algorithm is that it computes coresets that are small in size and subsets of the original data. We benchmark our algorithm for quality, efficiency, and scalability using synthetic data. We then apply our algorithm for computing LSA on the entire Wikipedia – a computation task hitherto not possible with state of the art algorithms. We see this work as a theoretical foundation and practical toolbox for a range of dimensionality reduction problems, and we believe that our algorithms will be used to

construct many other coresets in the future. Our project codebase is open-sourced and can be found here: `http://people.csail.mit.edu/mikhail/NIPS2016`.

## Footnotes

[1]Support for this research has been provided by Hon Hai/Foxconn Technology Group and NSFSaTC-BSF CNC 1526815, and in part by the Singapore MIT Alliance on Research and Technology through the Future of Urban Mobility project and by Toyota Research Institute (TRI). TRI provided funds to assist the authors with their research but this article solely reflects the opinions and conclusions of its authors and not TRI or any other Toyota entity. We are grateful for this support.

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
