[Supplementary Material]

# Supplementary Material

## A  Proof of Lemma 1

**Lemma 1.** *Let $z \in D^n$ be a distribution over $n$ unit vectors $a_1, \cdots, a_n$ in $\mathbb{R}^d$. For $\varepsilon \in (0,1)$, a sparse weight vector $w \in D^n$ of sparsity $s \leq 1/\varepsilon^2$ can be computed in $O(nd/\varepsilon^2)$ time such that*

$$\left\| \sum_{i=1}^n z_i \cdot a_i - \sum_{i=2}^n w_i \, a_i \right\|_2 \leq \varepsilon. \tag{9}$$

We note that the Caratheodory Theorem [4] proves Lemma 1 for the special case $\varepsilon = 0$ using only $d+1$ points. Our approach and algorithm can thus be considered as an $\varepsilon$-approximation for the Caratheodory Theorem, to get coresets of size independent of $d$. Note that our Frank-Wolfe-style algorithm might run more than $d+1$ or $n$ iterations without getting zero error, since the same point may be selected in several iterations. Computing in each iteration the closest point to the origin that is *spanned* by all the points selected in the previous iterations, would guarantee coresets of size at most $d+1$, and fewer iterations. Of course, the computation time of each iteration will also be much slower. '

*Proof.* We assume that $\sum_i z_i a_i = \mathbf{0}$, otherwise we subtract $\sum_j z_j a_j$ from each input vector $a_i$. We also assume $\varepsilon < 1$, otherwise the claim is trivial for $w = \mathbf{0}$. Let $w \in D^n$ such that $\|w\|_0 = 1$, and denote the current mean approximation by $c = \sum_i w_i a_i$. Hence, $\|c\|_2 = \|a_i\| = 1$.

The following iterative algorithm updates $c$ in the end of each iteration until $\|c\|_2 < \varepsilon$. In the beginning of the $N$th iteration the squared distance from $c$ to the mean (origin) is

$$\|c\|_2^2 \in [\varepsilon, \frac{1}{N}]. \tag{10}$$

The average distance to $c$ is thus

$$\sum_i z_i \|a_i - c\|_2^2 = \sum_i z_i \|a_i\|_2^2 + 2c^T \sum_i z_i a_i + \sum_i z_i \|c\|_2^2 = 1 + \|c\|_2^2 \geq 1 + \varepsilon \, ,$$

where the sum here and in the rest of the proof are over $[n]$. Hence there must be a $j \in [n]$ such that

$$\|q_j - c\|_2^2 \geq 1 + \varepsilon. \tag{11}$$

Let $r$ be the point on the segment between $a_j$ and $c$ at a distance $\rho := 1/\|a_j - c\|_2$ from $a_j$. Since $\|a_j - r\|_2 = \rho = \rho\|a_j - \mathbf{0}\|_2$, and $\|a_j - \mathbf{0}\|_2 = 1 = \rho\|a_j - c\|_2$, and $\angle(\mathbf{0}, a_j, c) = \angle(c, a_j, \mathbf{0})$, the triangle whose vertices are $a_j$, $r$ and $\mathbf{0}$ is similar to the triangle whose vertices are $a_j$, $\mathbf{0}$, and $c$ with a scaling factor of $\rho$. Therefore,

$$\|r - \mathbf{0}\|_2 = \rho \cdot \|\mathbf{0} - c\|_2 = \frac{\|c\|_2}{\|q_j - c\|_2}. \tag{12}$$

From (11) and (12), by letting $c'$ be the closest point to $\mathbf{0}$ on the segment between $a_j$ and $c$, we obtain

$$\|c'\|_2^2 \leq \|r\|_2^2 = \frac{\|c\|_2^2}{\|a_j - c\|_2^2} \leq \frac{\|c\|_2^2}{1 + \varepsilon}.$$

Combining this with (10) yields

$$\|c'\|_2^2 \leq \frac{\frac{1}{N}}{1 + \varepsilon} \leq \frac{\frac{1}{N}}{1 + \frac{1}{N}} = \frac{1}{N + 1}.$$

Since $c'$ is a convex combination of $a_j$ and $c$, there is $\alpha \in [0,1]$, such that $c' = \alpha a_j + (1 - \alpha)c$. Therefore,

$$c' = \alpha a_j + (1 - \alpha) \sum_i w_i a_i$$

and thus we have $c' = \sum_i w'_i a_i$, where $w' = (1 - \alpha)w + \alpha e_j$, and $e_j \in D^n$ is the $j$th standard vector. Hence, $\|w'\|_0 = N + 1$. If $\|c'\|_2^2 < \varepsilon$ the algorithm returns $c'$. Otherwise

$$\|c'\|_2^2 \in [\varepsilon, \frac{1}{N + 1}] \tag{13}$$

We can repeat the procedure in (10) with $c'$ instead of $c$ and $N + 1$ instead of $N$. By (29) $N + 1 \leq 1/\varepsilon$ so the algorithm ends after $N \leq 1/\varepsilon$ iterations. After the last iteration we return the center $c' = \sum_{i=1}^n w'_i a_i$ so

$$\left\| \sum_i (z_i - w'_i)a_i \right\|_2^2 = \|c'\|_2^2 \leq \frac{1}{N + 1} \leq \varepsilon.$$

$\square$

## B  Proof of Theorem 3

**Theorem 3** (Coreset for Low rank approximation). *For every $X \in \mathbb{R}^{d \times (d-k)}$ such that $X^T X = I$,*

$$\left| 1 - \frac{\|WAX\|^2}{\|AX\|^2} \right| \leq 5 \left\| \sum_{i=1}^{n} v_i v_i^T - W_{i,i} v_i v_i^T \right\|. \tag{14}$$

*Proof of Theorem 3.* Let $\varepsilon = \| \sum_{i=1}^{n} (1 - W_{i,i}^2) v_i v_i^T \|$. For every $i \in [n]$ let $t_i = 1 - W_{i,i}^2$. Set $X \in \mathbb{R}^{d \times (d-k)}$ such that $X^T X = I$. Without loss of generality we assume $V^T = I$, i.e. $A = U\Sigma$, otherwise we replace $X$ by $V^T X$. It thus suffices to prove that

$$\left| \sum_i t_i \|A_{i,:} X\|^2 \right| \leq 5\varepsilon \|AX\|^2. \tag{15}$$

Using the triangle inequality, we get

$$\left| \sum_i t_i \|A_{i,:} X\|^2 \right| \leq \left| \sum_i t_i \|A_{i,:} X\|^2 - \sum_i t_i \|(A_{i,1:k}, \mathbf{0}) X\|^2 \right| \tag{16}$$

$$+ \left| \sum_i t_i \|(A_{i,1:k}, \mathbf{0}) X\|^2 \right|. \tag{17}$$

We complete the proof by deriving bounds on (16) and (17).

**Bound on** (16): It was proven in [1] that for every pair of $k$-subspaces $S_1, S_2$ in $\mathbb{R}^d$ there is $u \geq 0$ and a $(k-1)$-subspace $T \subseteq S_1$ such that the distance from every point $p \in S_1$ to $S_2$ equals to its distance to $T$ multiplied by $u$. By letting $S_1$ denote the $k$-subspace that is spanned by the first $k$ standard vectors of $\mathbb{R}^d$, letting $S_2$ denote the $k$-subspace that is orthogonal to each column of $X$, and $y \in \mathbb{R}^k$ be a unit vector that is orthogonal to $T$, we obtain that for every row vector $p \in \mathbb{R}^k$,

$$\|(p, \mathbf{0}) X\|^2 = u^2 (py)^2. \tag{18}$$

After defining $x = \Sigma_{1:k,1:k} y / \|\Sigma_{1:k,1:k} y\|$, (16) is bounded by

$$\sum_i t_i \|(A_{i,1:k}, \mathbf{0}) X\|^2 = \sum_i t_i \cdot u^2 \|A_{i,1:k} y\|^2$$

$$= u^2 \sum_i t_i \|A_{i,1:k} y\|^2$$

$$= u^2 \sum_i t_i \|U_{i,1:k} \Sigma_{1:k,1:k} y\|^2$$

$$= u^2 \|\Sigma_{1:k,1:k} y\|^2 \sum_i t_i \|(U_{i,1:k}) x\|^2. \tag{19}$$

The left side of (19) is bounded by substituting $p = \Sigma_{j,1:k}$ in (18) for $j \in [k]$, as

$$u^2 \|\Sigma_{1:k,1:k} y\|^2 = \sum_{j=1}^{k} u^2 (\Sigma_{j,1:k} y)^2 = \sum_{j=1}^{k} \|(\Sigma_{j,1:k}, \mathbf{0}) X\|^2$$

$$= \sum_{j=1}^{k} \sigma_j^2 \|X_{j,:}\|^2 \leq \sum_{j=1}^{d} \sigma_d^2 \|X_{j,:}\|^2$$

$$= \|\Sigma X\|^2 = \|U\Sigma X\|^2 = \|AX\|^2. \tag{20}$$

The right hand side of (19) is bounded by

$$\left| \sum_i t_i \|(U_{i,1:k}) x\|^2 \right| = \left| \sum_i t_i (U_{i,1:k})^T U_{i,1:k} \cdot xx^T \right| = \left| xx^T \cdot \sum_i t_i (U_{i,1:k})^T U_{i,1:k} \right|$$

$$\leq \|xx^T\| \cdot \|\sum_i t_i (U_{i,1:k})^T U_{i,1:k}\| \tag{21}$$

$$\leq \|\sum_i t_i (v_{i,1:k})^T v_{i,1:k}\| \leq \|\sum_i t_i v_i^T v_i\| = \varepsilon \tag{22}$$

where (21) is by the Cauchy-Schwartz inequality and the fact that $\|xx^T\| = \|x\|^2 = 1$, and in (22) we used the assumption $A_{i,j} = U_{i,j}\sigma_j = v_{i,j}$ for every $j \in [k]$.

Plugging (20) and (22) in (19) bounds (16) as

$$|\sum_i t_i\|(A_{i,1:k},\mathbf{0})X\|^2| \le \varepsilon\|AX\|^2. \tag{23}$$

**Bound on** (17)**:** For every $i \in [n]$ we have

$$\|A_{i,:}X\|^2 - \|(A_{i,1:k},\mathbf{0})X\|^2$$
$$= 2(A_{i,1:k},\mathbf{0})XX^T(\mathbf{0},A_{i,k+1:d})^T + \|(\mathbf{0},A_{i,k+1:d})X\|^2$$
$$= 2A_{i,1:k}X_{1:k,:}(X_{k+1:d,:})^T(A_{i,k+1:d})^T + \|(\mathbf{0},A_{i,k+1:d})X\|^2$$
$$= 2\sum_{j=1}^k A_{i,j}X_{j,:}(X_{k+1:d,:})^T(A_{i,k+1:d})^T + \|(\mathbf{0},A_{i,k+1:d})X\|^2$$
$$= \sum_{j=1}^k 2\sigma_j X_{j,:}(X_{k+1:d,:})^T \cdot \|\sigma_{k+1:d}\|v_{i,j}(v_{i,k+1:d})^T +$$
$$\|\sigma_{k+1:d}\|^2\|(\mathbf{0},v_{i,k+1:d})X\|^2. \tag{24}$$

Summing this over $i \in [n]$ with multiplicative weight $t_i$ and using the triangle inequality, will bound (17) by

$$\left|\sum_i t_i\|A_{i,:}X\|^2 - \sum_i t_i\|(A_{i,1:k},\mathbf{0})X\|^2\right|$$

$$\le \left|\sum_i t_i\sum_{j=1}^k 2\sigma_j X_{j,:}(X_{k+1:d,:})^T \right. \tag{25}$$

$$\left. \cdot \|\sigma_{k+1:d}\|v_{i,j}(v_{i,k+1:d})^T\right|$$

$$+ \left|\sum_i t_i\|\sigma_{k+1:d}\|^2\|(\mathbf{0},v_{i,k+1:d})X\|^2\right|. \tag{26}$$

The right hand side of (25) is bounded by

$$\left|\sum_{j=1}^k 2\sigma_j X_{j,:}(X_{k+1:d})^T \cdot \|\sigma_{k+1:d}\|\sum_i t_i v_{i,j}(v_{i,k+1:d})^T\right|$$

$$\le \sum_{j=1}^k 2\sigma_j\|X_{j,:}X_{k+1:d}\| \cdot \|\sigma_{k+1:d}\|\|\sum_i t_i v_{i,j}v_{i,k+1:d}\| \tag{27}$$

$$\le \sum_{j=1}^k (\varepsilon\sigma_j^2\|X_{j,:}\|^2 + \frac{\|\sigma_{k+1:d}\|^2}{\varepsilon}\|\sum_i t_i v_{i,j}v_{i,k+1:d}\|^2) \tag{28}$$

$$\le 2\varepsilon\|AX\|^2, \tag{29}$$

where (27) is by the Cauchy-Schwartz inequality, (28) is by the inequality $2ab \le a^2 + b^2$. In (29) we used the fact that $\sum_i t_i(v_{i,1:k})^T v_{i,k+1:d}$ is a block in the matrix $\sum_i t_i v_i v_i^T$, and

$$\|\sigma_{k+1:d}\|^2 \le \|AX\|^2 \quad \text{and} \quad \sum_{j=1}^k \sigma_j^2\|X_{j,:}\|^2 \tag{30}$$
$$= \|\Sigma_{1:k,1:k}X_{1:k,:}\|^2 \le \|\Sigma X\|^2 \le \|AX\|^2.$$

Next, we bound (26). Let $Y \in \mathbb{R}^{d\times k}$ such that $Y^T Y = I$ and $Y^T X = \mathbf{0}$. Hence, the columns of $Y$ span the $k$-subspace that is orthogonal to each of the $(d-k)$ columns of $X$. By using the Pythagorean Theorem and

then the triangle inequality,

$$\|\sigma_{k+1:d}\|^2 |\sum_i t_i\|(\mathbf{0}, v_{i,k+1:d})X\|^2| \tag{31}$$

$$=\|\sigma_{k+1:d}\|^2 |\sum_i t_i\|(\mathbf{0}, v_{i,k+1:d})\|^2$$

$$-\sum_i t_i\|(\mathbf{0}, v_{i,k+1:d})Y\|^2|$$

$$\leq \|\sigma_{k+1:d}\|^2 |\sum_i t_i\|v_{i,k+1:d}\|^2| \tag{32}$$

$$+\|\sigma_{k+1:d}\|^2 |\sum_i t_i\|(\mathbf{0}, v_{i,k+1:d})Y\|^2|. \tag{33}$$

For bounding (33), observe that $Y$ corresponds to a $(d-k)$ subspace, and $(\mathbf{0}, v_{i,k+1:d})$ is contained in the $(d-k)$ subspace that is spanned by the last $(d-k)$ standard vectors. Using same observations as above (18), there is a unit vector $y \in \mathbb{R}^{d-k}$ such that for every $i \in [n]$ $\|(\mathbf{0}, v_{i,k+1:d})Y\|^2 = \|(v_{i,k+1:d})y\|^2$. Summing this over $t_i$ yields,

$$|\sum_i t_i\|(\mathbf{0}, v_{i,k+1:d})Y\|^2| = |\sum_i t_i\|v_{i,k+1:d}y\|^2|$$

$$= |\sum_i t_i \sum_{j=k+1}^d v_{i,j}^2 y_{j-k}^2| = |\sum_{j=k+1}^d y_{j-k}^2 \sum_i t_i v_{i,j}^2|.$$

Replacing (33) in (31) by the last inequality yields

$$\|\sigma_{k+1:d}\|^2 |\sum_i t_i\|(\mathbf{0}, v_{i,k+1:d})X\|^2|$$

$$\leq \|\sigma_{k+1:d}\|^2 (|\sum_i t_i v_{i,d+1}^2| + \sum_{j=k+1}^d y_{j-k}^2 \|\sum_i t_i v_i v_i^T\|) \tag{34}$$

$$\leq \|\sigma_{k+1:d}\|^2 (\varepsilon + \varepsilon \sum_{j=k+1}^d y_{j-k}^2) \leq 2\varepsilon\|AX\|^2, \tag{35}$$

where (34) follows since $\sum_i t_i v_{i,j}^2$ is an entry in the matrix $\sum_i t_i v_i v_i^T$, in (35) we used (30) and the fact that $\|y\|^2 = 1$. Plugging (29) in (25) and (35) in(20) gives the desired bound on (17) as

$$|\sum_i t_i\|A_{i,:}X\|^2 - \sum_i t_i\|(A_{i,1:k}, \mathbf{0})X\|^2| \leq 4\varepsilon\|AX\|^2.$$

Finally, using (23) in (16) and the last inequality in (17), proves the desired bound of (15).

$\square$

## C    Analysis of Algorithm 1

Algorithm 1 contains the full listing of the construction algorithm for the coreset for sum of vectors.

**Input:** $A$: $n$ input points $a_1, \ldots, a_n$ in $\mathbb{R}^d$; $\varepsilon > 0$: the nominal approximation error.

**Output:** a non-negative vector $w \in [0, \infty)^n$ of only $O(1/\varepsilon^2)$ non-zeros entries which are the non-negative weights of the corresponding points selected for the coreset.

**Analysis:** The first step is to translate and scale the input points such that the mean is zero and the variance is 1 (lines 4–5). After initialization (lines 6–8), we begin the main iterative steps of the algorithm. First we find the index $j$ of the farthest point from the initial point $a_1$. The next point added to the coreset is denoted by $p = a_j$. Next we compute $\|c - p\|$, the distance from the current point $p$ to the previous center $c$. In order to do this we compute $G = W' \cdot A_J$ where $J$ is the set of all previously added indices $j$, starting with the first point, and $W'$ is defined in line 11. Note that $G$ also gives us the error of the current iteration, $\varepsilon = \text{trace}(G\,G^T)$ (line 23). Next we find the point $c'$ on the line from $c$ to $p$ that is closest to the origin, and find the distance between the current center $c$ and the new center $c'$ (lines 12–16). Finally, the ratio of distances between the current center, farthest point, and new center give us a value for $\alpha$, the amount by which we update the coreset weights (lines 17–20).

---

**Algorithm 1** CORESET-SUMVECS$(A, \varepsilon)$

---

1: **Input:** $A$: $n$ input points $a_1, \ldots, a_n$ in $\mathbb{R}^d$
2: **Input:** $\varepsilon \in (0, 1)$: the approximation error
3: **Output:** $w \in [0, \infty)^n$: non-negative weights
4: $A \leftarrow A - \text{mean}(A)$
5: $A \leftarrow c\,A$ where $c$ is a constant s.t. $\text{var}(A) = 1$
6: $w \leftarrow (1, 0, \ldots, 0)$
7: $j \leftarrow 1, \; p \leftarrow A_j, \; J \leftarrow \{j\}$
8: $M_j = \left\{ y^2 \mid y = A \cdot A_j^T \right\}$
9: **for** $i = 1, \ldots, n$ **do**
10: $\quad j \leftarrow \text{argmin}\,\{w_J \cdot M_J\}$
11: $\quad G \leftarrow W' \cdot A_J$ where $W'_{i,i} = \sqrt{w_i}$
12: $\quad \|c\| = \|G^T G\|_F^2$
13: $\quad c \cdot p = \sum_{i=1}^{|J|} G\,p^T$
14: $\quad \|c - p\| = \sqrt{1 + \|c\|^2 - c \cdot p}$
15: $\quad \text{comp}_p(v) = 1/\|c - p\| - (c \cdot p)\,/\|c - p\|$
16: $\quad \|c - c'\| = \|c - p\| - \text{comp}_p(v)$
17: $\quad \alpha = \|c - c'\|/\|c - p\|$
18: $\quad w \leftarrow w(1 - |\alpha|)$
19: $\quad w_j \leftarrow w_j + \alpha$
20: $\quad w \leftarrow w/\sum_{i=1}^{n} w_i$
21: $\quad M_j \leftarrow \left\{ y^2 \mid y = A \cdot A_j^T \right\}$
22: $\quad J \leftarrow J \cup \{j\}$
23: $\quad$ **if** $\|c\|^2 \leq \varepsilon$ **then**
24: $\quad\quad$ **break**
25: $\quad$ **end if**
26: **end for**
27: **return** $w$

---

The algorithm then updates the recorded indices $J$, update the lookup table $M$ of previously computed row inner products for subsequent iterations, and repeat lines 10–26 until the loop terminates. The terminating conditions depend on the system specification – we may wish to bound the error, or the number of iterations. Moreover, if the update value $\alpha$ is below a specified threshold, we may also terminate the loop if such threshold is lower than a desired level of accuracy.

## D  Analysis of Algorithm 2

Algorithm 2 contains the full listing of the construction algorithm for the coreset for low rank approximation.

**Input:** $A$: $n$ input points $a_1, \ldots, a_n$ in $\mathbb{R}^d$; $k \geq 1$: the approximation rank; $\varepsilon > 0$: the nominal approximation error.

**Output:** a non-negative vector $w \in [0, \infty)^n$ of only $O(1/\varepsilon^2)$ non-zeros entries which are the non-negative weights of the corresponding points selected for the coreset.

**Analysis:** Algorithm 2 starts by computing the $k$-SVD of input matrix $A$ (line 5). This is possible because we use the streaming model, so that the input arrives in small blocks. For each block we perform the computation to create its coreset. By merging the resulting coresets we preserve sparsity and can aggregate the coreset for $A$. Lines 7–8 use the $k$-SVD of this small input block to restructure the input matrix $A$ into a combination of the columns of $A$ corresponding to its $k$ largest eigenvalues and the remaining columns of $D$, the singular values of $A$.

After initialization, we begin the main iterative steps of the algorithm. Note that lines 12–19 of Algorithm 2 are heavily optimized but functionally equivalent to lines 9–27 of Algorithm 1 – the end result in both cases is a computation of $\alpha$ at each iteration of the for loop, and an update to the vector of weights $w$. First we find the index $j$ of the farthest point from the initial point $a_1$ (Line 13). The next point is implicitly added to the coreset is by updating $w$, and in turn affects the next farthest point as the computation $wXX_i$ is performed iteratively. The variables $a, b, c$ implicitly compute the distance from the current point $p$ to the previous center $q$, the error of the current iteration $\varepsilon$, the point on the line from the $p$ to $q$ that is closest to the origin, and the distance between the current center $q$ and the new center $q'$. Finally, line 17 updates $\alpha$ and line 18 updates $w$ using the new value of $\alpha$.

---

**Algorithm 2** CORESET-LOWRANK$(A, k, \varepsilon)$

---

1: **Input:** $A$: A sparse $n \times d$ matrix
2: **Input:** $k \in \mathbb{Z}_{>0}$: the approximation rank
3: **Input:** $\varepsilon \in \left(0, \frac{1}{2}\right)$: the approximation error
4: **Output:** $w \in [0, \infty)^n$: non-negative weights
5: Compute $U\Sigma V^T = A$, the SVD of $A$
6: $R \leftarrow \Sigma_{k+1:d, k+1:d}$
7: $P \leftarrow$ matrix whose $i$-th row $\forall i \in [n]$ is
8: $\quad P_i = (U_{i,1:k}, U_{i,k+1:d} \cdot \frac{R}{\|R\|_F})$
9: $X \leftarrow$ matrix whose $i$-th row $\forall i \in [n]$ is
10: $\quad X_i = P_i / \|P_i\|_F$
11: $w \leftarrow (1, 0, \ldots, 0)$
12: **for** $i = 1, \ldots, \lceil k^2/\varepsilon^2 \rceil$ **do**
13: $\quad j \leftarrow \mathrm{argmin}_{i=1,\ldots,n}\{wXX_i\}$
14: $\quad a = \sum_{i=1}^n w_i (X_i^T X_j)^2$
15: $\quad b = \dfrac{1 - \|PX_j\|_F^2 + \sum_{i=1}^n w_i \|PX_i\|_F^2}{\|P\|_F^2}$
16: $\quad c = \|wX\|_F^2$
17: $\quad \alpha = (1 - a + b) / (1 + c - 2a)$
18: $\quad w \leftarrow (1 - \alpha)I_j + \alpha w$
19: **end for**
20: **return** $w$

---

The algorithm terminates after $k^2/\varepsilon^2$ iterations, and we omit the explicit computation of $\varepsilon$ since it is implied in the guarantees proven in the following section. As in Algorithm 1, the terminating conditions depend on the system specifications. We may wish to bound the error, or the number of iterations, or the update value $\alpha$.

## E  Experimental Results – Synthetic Data

Synthetic data provides us with a ground-truth to objectively evaluate the quality, efficiency, and scalability of our system.

**Approximation error.** We carried out experiments on a moderate size sparse input of $(5000 \times 1000)$ to evaluate the relationship between the error $\varepsilon$ and the number of iterations of the algorithm $N$. for a hyperplane coreset (i.e. $k = d - 1$). Fig. 1d shows how the characteristic function of the approximation error $f(N)$ behaves with respect to increasing number of iterations $N$ (normalized to $N = n$). Note that three of the plotted functions $f(N)$ converge as $N$ increases, while the last one ramps up and then increases linearly. From this we conclude that $\varepsilon$ decreases at a true rate somewhere between the rates of increase of $f(N) = N \log N$ and $f(N) = N^2$. The true characteristic $f^*(N) + C$ indicates the theoretical breakpoint between increasing and decreasing error.

We then compare our coreset against uniform sampling and weighted random sampling, using the squared norms of $U$ $(A = U\Sigma V^T)$ as the weights. Tests were carried out on a small subset of Wikipedia ($n = 1000$, $d = 257K$) to ensure representative data structure. Figure 1a–1c shows the results. As expected, approximation error decreases with coreset size, as well as the subspace rank. (Note that since our algorithm is deterministic, there is zero variance in the approximation error.)

**Running time.** We evaluate the efficiency of our algorithm by comparing the running time (coreset construction) against the built-in MATLAB `svds` function. Fig. 2a shows the runtimes of our coreset compared against MATLAB svds. We used a fixed dimensionality $d = 1000$, approximation rank $k = 100$, sparsity $10^{-6}$ and evaluated construction time for increasing input size $N$. The results are plotted as a function of the log of the input size to show the order of magnitude difference in performance.

Besides the fact that our algorithm minimizes the Frobenius norm and support PCA, an important advantage of our technique compared to existing coreset constructions is that it is much numerically stable and faster in practice. For example, the result of [7] is based on the technique of [3]. This technique needs to compute many inverse of matrices during the computation, which makes it not only less stable but also very inefficient.

## F  Experimental Results – Latent Semantic Analysis of Wikipedia

For these experiments we used three types of machines:

**A[10000x100000], sparsity=0.033**

(a) Relative error $(k = 10)$

Figure 2: Fig. 2a shows the runtimes of our coreset compared against MATLAB svds.

1. Regular desktop computer with quad-core Intel Xeon E5640 CPU @2.67GHz, 6GB RAM (low spec).

2. Modern laptop with quad-core Intel i7-4500U CPU @1.8GHz, 16GB RAM (medium spec).

3. High-performance computing clusters on Amazon Web Services (AWS) as well as local clusters, e.g. an EC2 c3.8xlarge machine with 32-core Intel Xeon E5-2680v2 vCPU @2.8Ghz, 60GB RAM (high spec).

We compute the coreset using a buffer stream of size $N/2$, parallelized across 64 nodes on Amazon Web Services (AWS) clusters. The 64 individual coresets are then unified into a single coreset. Figure 1e shows the running time of our algorithm compared against svds for increasing dimensionality $d$ and a fixed input size $n = 3.69M$ (number of documents). Note that this is a log-scale plot of dimensionality against running time, so the differences in performance represent orders of magnitude. The desktop computer with 6GB RAM crashed for $d = 2000$ and was omitted from the plot. The same algorithm running on the cluster (blue plot) outperformed the laptop (red plot), which also quickly ran out of memory. Comparing svds computation on AWS against our coreset (green plot) highlights the difference in performance for identical computer architectures. As the dimensionality $d$ increases, any algorithm dependent on $d$ will eventually crash, given a large enough input.

We show that our coreset can be used to create a topic model of $k = 100$ topics for the entire English Wikipedia, with a fixed memory requirement and coreset size of just $N = 1000$ words. We compute the projection of the coresets on a subspace of rank $k$ to generate the topics. Table 1 shows a selection of 10 of the most highly weighted words from 4 of the computed topics. The total running time, including coreset construction, merging and topic extraction was 140.66 min.

A cursory glance at the words suggests that the "themes" of these topics are (1) urban planning, (2) economy and finance, (3) road safety, (4) entertainment. This serves as a qualitative proof of concept that our system can produce meaningful results topics on very large datasets. We view this result optimistically, as proof of concept that our system can be used to compute a topic model of the English language. A more objective analysis would involve using a corpus of tagged documents as a ground truth, projecting the corresponding vectors onto our topics, and comparing the classification error against topics computed by other systems. This is the subject of our ongoing work.

| Topic 1 | Topic 2 | Topic 3 | Topic 4 |
|---------|---------|---------|---------|
| US | credit | drivers | comedy |
| highway | risk | distracted | nominated |
| bridge | plan | phone | actress |
| road | union | driver | awards |
| river | interest | text | television |
| traffic | rating | car | episode |
| downtown | earnings | brain | musical |
| bus | capital | accidents | writing |
| harbor | liquidity | visual | tv |
| street | asset | crash | directing |
| . . . | . . . | . . . | . . . |

Table 1: Example of the highest-weighted words from 4 topics of the $k = 100$ topic model of Wikipedia computed by our algorithm