[Reviews · NeurIPS 2016]

Reviewer 1

Summary

The paper studies the problem of selecting a small number of rows and reweightings of them so as the approximate the sum of squared distances of the original point set to any k-dimensional affine space. The authors show O(k^2/eps^2) rows can be chosen deterministically. The base case is showing it for k = 0, and the method seems something like a derandomization of Cheybshev's inequality.

Qualitative Assessment

This paper makes some pretty critical mistakes regarding previous work. For one, they cite [8], but they should in fact be citing Cohen et al. "Dimensionality Reduction for k-Means Clustering and Low Rank Approximation" This is not just a typo - the authors go on to state a result of [8] about operator norm rather than the result of the Cohen et al. paper - namely, the Cohen et al. paper achieves O(k/eps^2) rescaled columns deterministically for exactly the same problem considered in this submission - see part 5 of Lemma 11 and section 7.3 based on BSS. This is much stronger than the O(k^2/eps^2) rescaled columns achieved in the submission. This directly contradicts their sentence "Our main result is the first algorithm for computing an (k,eps)-coreset C of size independent of both n and d". The authors also say later [8,7] minimize the 2-norm - [8] is the wrong reference again! Use the Cohen et al. paper above. The authors also don't compare their work to the result in Theorem 8 here: http://arxiv.org/pdf/1511.07263v1.pdf I'm also not entirely convinced about their comparison to [7], that is, it's not clear to me [7] couldn't be made deterministic and hold for rank-k approximation using similar techniques. Although the above are the negative points, I do like that the paper does deterministic sampling without BSS, which could be, as the authors show in the experiments, more practical than BSS. This was the most interesting part for me, even though it gets a worse k^2/eps^2 bound. Because of the deterministic argument without BSS, I think the paper can be accepted, provided a more proper comparison is made to the above work and that the paper focuses more on this practical aspect.

Confidence in this Review

2-Confident (read it all; understood it all reasonably well)


Reviewer 2

Summary

This paper studies a dimension reduction problem of finding a (weighted) subset of n vectors to approximate the sum of squared distances from those n vectors to any other k-dimensional affine subspace. which can be seen as an "online" PCA problem. Their main contribution is to first prove the existence of such subset of size independent of the number of vectors n and the dimensionality of these vectors d. And they give a computationally efficient algorithm to compute it. They also show the application of their algorithm on the latent semantic analysis of English Wikipedia.

Qualitative Assessment

This is an interesting paper and the result is novel. They find a corset of size independent of the input dimension, which could significantly reduce the computation time of dimension reduction problem. However, there seems to be an error in the proof. To me, theorem 2 is the key step to establishing the main result. But I do not see why we can replace v_i (a vector) by v_i v_i^T (a matrix) in using Theorem 2. (line 265). Also, it is better to provide more intuitions on how to reduce the running time of Algorithm 1 in proving Theorem 1. I think the result of this paper could be improved by providing a lower bound. Is it possible to construct a special input matrix such that any corset of size o(k^2/\epsilon^2) can not achieve 1+\epsilon approximation? The organization of this paper is very clear and easy to follow, though there are some typos. The authors sometimes use || || to denote the operator norm of a matrix, sometimes use it as the vector l_2 norm (Theorem 2, line 250-252). And there seems to be a typo on line 267. Do the authors mean equation (5)? The variance is missing on the line 5 of Algorithm 1.

Confidence in this Review

2-Confident (read it all; understood it all reasonably well)


Reviewer 3

Summary

In this paper, a coreset based method is proposed for feature selection (dimensionality reduction) for very large scale sparse matrices. An efficient algorithm is exploited to compute the coreset with theoretical analysis on the bounds of size and running time. Experimental results illustrate the effectiveness and superiority of the proposed methods.

Qualitative Assessment

In this paper, a coreset based method is proposed for feature selection (dimensionality reduction) for very large scale sparse matrices. An efficient algorithm is exploited to compute the coreset with theoretical analysis on the bounds of size and running time. Experimental results illustrate the effectiveness and superiority of the proposed methods. The paper is well written and organized. However, some types should be mentioned. For example, In Definition 1, the constrain on w_i should be consistent.

Confidence in this Review

1-Less confident (might not have understood significant parts)


Reviewer 4

Summary

This paper addresses the computation of PCA for extremely large sparse matrices. In particular, the authors use coresets, which are a weighted subset of rows of the data matrix. This paper shows how to compute a coreset that is independent of the dimensions of the data matrix. Moreover, they apply the algorithm to the very large Wikipedia document-term matrix, which is an impressive demonstration.

Qualitative Assessment

The paper is well written, and the motivation is both clear and compelling. Moreover, the concept of coresets and their application to a very challenging problem (the wikipedia dataset) is impressive. Overall, I enjoyed reading this paper and it made me think.

Confidence in this Review

2-Confident (read it all; understood it all reasonably well)


Reviewer 5

Summary

This paper presented a practical solution with performance guarantees to the problem of dimensionality reduction for very large scale sparse matrices, which uses a weighted subset of the data, and is independent of both the size and dimensionality of the data. Furthermore, an efficient algorithm for computing such a reduction, with provable bounds on size and running time. Also, a system that implements this dimensionality reduction algorithm and an application of the system to compute latent semantic analysis (LSA) of the entire English Wikipedia. It is an interesting work. Currently there are lots of algorithms for large-scale data, and the authors should discuss with such work. The writing of this paper is hard to understand. The paper needs more language proof. There are a lot of typos. Also it is better to denote the meaning of symbols, e.g. || in equation (2)-(i,ii). Equation (3) is hard to understand, please clarify more. The distance between A and S is the same to manifold distance [R1]. The authors should cite this work. [R1] Wang, Ruiping, et al. "Manifold-manifold distance with application to face recognition based on image set." CVPR, 2008 The format of reference should be consistent. Please check carefully.

Qualitative Assessment

I think the paper has novelty but the language parts should be improved more.

Confidence in this Review

2-Confident (read it all; understood it all reasonably well)